# Discovery of Novel Andrographolide Derivatives as Antiviral Inhibitors against Human Enterovirus A71

**DOI:** 10.3390/ph15020115

**Published:** 2022-01-18

**Authors:** Jie Kai Tan, Ran Chen, Regina Ching Hua Lee, Feng Li, Kun Dai, Guo-Chun Zhou, Justin Jang Hann Chu

**Affiliations:** 1Laboratory of Molecular RNA Virology and Antiviral Strategies, Department of Microbiology and Immunology, Yong Loo Lin School of Medicine, National University of Singapore, Singapore 117545, Singapore; e0031885@u.nus.edu (J.K.T.); miclch@nus.edu.sg (R.C.H.L.); 2School of Pharmaceutical Sciences, Nanjing Tech University, Nanjing 211816, China; 202162118026@njtech.edu.cn (R.C.); fengli9203@163.com (F.L.); daikun@renfu.com.cn (K.D.); 3Infectious Disease Translational Research Programme, Yong Loo Lin School of Medicine, National University of Singapore, Singapore 117597, Singapore; 4Collaborative and Translation Unit for HFMD, Institute of Molecular and Cell Biology, Agency for Science, Technology and Research, Singapore 138673, Singapore

**Keywords:** human enterovirus A71, andrographolide, quinolinoxy, olefin, viral RNA replication, host-targeting, broad-spectrum anti-enterovirus agent

## Abstract

Hand-foot-and-mouth disease (HFMD) caused by human enterovirus A71 (EV-A71) infection has been associated with severe neurological complications. With the lack of an internationally approved antiviral, coupled with a surge in outbreaks globally, EV-A71 has emerged as a neurotropic virus of high clinical importance. Andrographolide has many pharmacological effects including antiviral activity and its derivative, andrographolide sulfonate, has been used in China clinically to treat EV-A71 infections. This study sought to identify novel andrographolide derivatives as EV-A71 inhibitors and elucidate their antiviral mode of action. Using an immunofluorescence-based phenotypic screen, we identified novel EV-A71 inhibitors from a 344-compound library of andrographolide derivatives and validated them with viral plaque assays. Among these hits, **ZAF-47**, a quinolinoxy-andrographolide, was selected for downstream mechanistic studies. It was found that **ZAF-47** acts on EV-A71 post-entry stages and inhibits EV-A71 protein expression. Subsequent luciferase studies confirm that **ZAF-47** targets EV-A71 genome RNA replication specifically. Unsuccessful attempts in generating resistant mutants led us to believe a host factor is likely to be involved which coincide with the finding that **ZAF-47** exhibits broad-spectrum antiviral activity against other enteroviruses (CV-A16, CV-A6, Echo7, CV-B5, CV-A24 and EV-D68). Furthermore, **ZAF-46** and **ZAF-47**, hits from the screen, were derivatives of the same series containing quinolinoxy and olefin modifications, suggesting that an andrographolide scaffold mounted with these unique moieties could be a potential anti-EV-A71/HFMD strategy.

## 1. Introduction

*Andrographis paniculata* (Burm.f.) Nees (family: Acanthaceae), also known as the “King of Bitters”, is a medicinal plant native to India and Sri Lanka and widely distributed in tropical and subtropical regions such as Southeast Asia and China [1]. It is extensively cultivated due to its high medicinal value which includes anti-atherosclerotic, antidiarrheal, antihyperglycemic, anti-inflammatory, antimicrobial, antioxidant, antiplatelet aggregation, choleretic, hepatoprotective and hypotensive activities [2,3]. Among the many constituents of diterpenoids, flavonoids and polyphenols contributing to its pharmacology, andrographolide (1) stands out as the most abundant and biologically active component [1,3]. Andrographolide is anticancer, anti-inflammatory, antibacterial, immunomodulatory [2] but more importantly, antiviral [4]. It is reported to be effective against chikungunya virus (CHIKV) [5], dengue virus (DENV) [6,7], Epstein–Barr virus (EBV) [8], influenza A virus (IAV) [9], hepatitis B virus (HBV) [10], hepatitis C virus (HCV) [11], human immunodeficiency virus (HIV) [12,13], herpes simplex virus (HSV) [14,15] and SARS-CoV-2 [16,17]. It is also effective against the *Enterovirus* genus like enterovirus D68 (EV-D68) [18].

Among the *Enterovirus* genus, human enterovirus A71 (EV-A71) is notoriously known for causing epidemics of hand-foot-and-mouth disease (HMFD) among young children. Initially isolated in the United States in 1969 [19], EV-A71 later spread to Australia [20], Bulgaria [21], Japan [22] and Hungary [23]. By the 2000s, EV-A71 became widespread in the Asia–Pacific Region. Countries and regions such as Malaysia [24], Taiwan [25,26,27,28], Japan [29], Singapore [30,31] and China [32] had been experiencing recurring EV-A71 outbreaks throughout the years. China, within a short span of 7 years from 2008 to 2015, had tallied more than 13 million cases which includes 3,322 deaths [32].

Transmitted primarily by oral–fecal route [33], HFMD is a mild and self-limiting disease that affects young children. Clinical symptoms of HFMD include blisters appearing on the palms and soles, ulcerations in the mouth and on the tongue and fever [34]. HFMD can be caused by different members in the *Enterovirus* genus and while it typically resolves in a few days, cases caused by EV-A71 can result in more severe neurological and cardiopulmonary complications such as aseptic meningitis, brainstem encephalitis, pulmonary oedema and even death [34]. For example, a 1998 EV-A71 outbreak in Taiwan reported 129,106 HMFD or herpangina cases with 405 patients experiencing severe neurological and cardiopulmonary complications resulting in 78 deaths [25]. Recently, EV-A71 outbreaks involving severe complications had been observed in the United States [35] and European countries including Spain [36] and France [37], suggesting that the virus may be gradually spreading out of the Asia–Pacific region to other countries.

Unfortunately, there is no existing vaccine or antiviral treatment approved internationally for use against EV-A71. Current treatment mainly focuses on symptomatic relief. With rising cases of EV-A71 globally and the severe disease outcome associated with the infection, there is an urgent need to develop effective and potent antivirals against EV-A71. As such, the ability of andrographolide to inhibit the *Enterovirus* genus piques our interest. We hypothesize that using andrographolide as a lead compound, we would be able to synthesize new derivatives that would be effective against EV-A71. Indeed, in China, an andrographolide sulfonate complex known as “Xiyanping” had already been adopted clinically to treat HFMD [38,39]. While the clinical efficacy of “Xiyanping” is mainly immunomodulatory [40], in this study we report novel andrographolide derivatives that have direct antiviral activity against EV-A71. Using andrographolide as a lead compound, we successfully synthesized 344 andrographolide derivatives and, upon an immunofluorescence-based high-throughput screen, we have identified 19 hits. Of these 19 compounds, **ZAF-46** and **ZAF-47** belonging to the same series exhibit potent anti-EV-A71 activity and low cytotoxicity, following which, **ZAF-47** was selected for downstream analyses to elucidate their possible antiviral mechanism of action.

## 2. Results

### 2.1. High Throughput Screening of Compound Library Reveals Novel EV-A71 Inhibitors

The screening assay was built using an immunofluorescence approach that was previously established and validated [41], with DAPI-stained nuclei accounting for total cell count and FITC-stained EV-A71 accounting for infected cell count upon EV-A71 infection and compound treatment. Both cell counts would then be used to calculate percentage infection and percentage inhibition to identify for hits. Before the screening platform can be employed to screen for potential antivirals, its robustness has to be appraised. In this study, Z-factor was calculated to assess the robustness of the screening platform. Firstly, seeded RD cells were either infected with EV-A71 (Multiplicity of Infection of 1) or mock-infected with media. The cells were then fixed, stained and imaged 12 h post-infection (hpi). The IFA images were analyzed with the Cell Profiler software to determine the infection rate of each well. Between the virus-infected and mock-infected wells, a clear distinction in infection rates could be observed (Figure 1a). The closer Z-factor is to 1, the greater the data quality and hence the robustness of the screening platform [42]. In this case, the Z-factor, measured at 0.818, was indicative of a highly robust screening platform for high-throughput screening. The constructed screening platform was then employed to screen the compound library made up of 344 chemically synthesized novel compounds analogous to andrographolide. The compounds were screened at 10 µM with IFA images taken and analyzed to identify for hits. Treatment with 0.1% DMSO was used as vehicle control. Hits were identified using two criteria: inhibition rate >50% and nuclei count comparable to the 0.1% DMSO treated wells. From the EV-A71 screen, 36 compounds showed >50% inhibition (Figure 1b) and 19 compounds were identified as hits (Table 1) and proceeded with validation studies.

### 2.2. Validation of EV-A71 Hits

Firstly, we performed cell viability assay to evaluate the cytotoxicity profile of the 19 hits. Cells seeded overnight were treated with the hit compounds at 10 µM. Out of the 19 hits, 4 hits showed cell viability higher than 80% (Figure 2a). Anti-EV-A71 activity of AGP-253 had been illustrated and discussed previously [43]. We were particularly interested in **ZAF-46** and **ZAF-47** as they belonged to the same series bearing novel 14-quinolinoxy and 17-hydro-8,9-olefin modifications (Figure 2b) that could potentially open up new targets for EV-A71 therapeutics. Additional cell viability assays were performed for **ZAF-46** and **ZAF-47** at different concentrations to identify the low cytotoxic concentrations for downstream post-treatment assays. For post-treatment assays, cells seeded overnight were infected with EV-A71 and subsequently treated with **ZAF-46** and **ZAF-47** at different concentrations. We found that treatment with either **ZAF-46** or **ZAF-47** led to a significant dose-dependent reduction in viral titres. Treatment with **ZAF-46** led to 1.4 log reduction at 10 µM and 2.4 log reduction at 20 µM (Figure 2c) while treatment with **ZAF-47** led to 0.8 log reduction at 5 µM, 1.1 log reduction at 10 µM and 2.1 log reduction at 20 µM (Figure 2d). CC_50_ and IC_50_ for both compounds were determined and used to derive their SI (Table 2). **ZAF-47** was selected for further downstream analysis due to its higher SI value which indicate higher pharmacological value for clinical use.

### 2.3. Synthesis of **ZAF-46** and **ZAF-47**

As shown in Figure 1, intermediate **2** was prepared according to the reported references [44,45] by starting from andrographolide (**1**), and then the synthetic process from **2** to the 8,9-olifen **6** was adopted from our recently published method [46]. Briefly, acylation of **2** at 14-OH by *p*-nitrobenzoyl chloride provided compound **3**, the olefin migration step of 8,17-olefin (**3**) to 8,9-olefin (**4**) was conducted by the treatment of 85% H_3_PO_4_ followed by 3,19-protection of **4** by 2,2-dimethoxypropane and PPTS yielded 3,19-acetonylidene **5,** de-acetylation at 14-position of **5** in methanol by Lithium carbonate afforded the key intermediate **6**. After Mitsunobu reaction of **6** at 14-position was conducted to give **7,** de-protection of 3,19-acetonylidene by *p*-TSA in methanol obtained diol **8**, selectively acetylation by acetyl chloride of primary alcohol of 19-position afforded **ZAF-46** and oxidation by DMP of 3-alcohol to 3-ketone produced **ZAF-47**.

### 2.4. **ZAF-47** Targets EV-A71 Post Entry Stages

To determine the time window in the EV-A71 replication cycle in which the inhibitory activity of **ZAF-47** remains active, we performed time-of-removal (TOR) and time-of-addition (TOA) assays. **ZAF-47** was either removed from or added on the infected cells at fixed time points and upon 12 hpi, all supernatants were collected to quantify the viral titres. TOR assay would enable us in pinpointing the time point whereby the removal of **ZAF-47** would not remove any inhibitory effect and a plateauing of viral titre would be observed, whereas TOA assay would enable us in pinpointing the time point whereby the addition of **ZAF-47** would no longer have any inhibitory effect and an increase in viral titre would be observed. Combining both assays, it was observed that plateauing of viral titre began after **ZAF-47** was removed 4 hpi (Figure 3a) and only increased after **ZAF-47** was added 6 hpi (Figure 3b). This suggests that the antiviral activity of **ZAF-47** is likely to act between 4 and 6 hpi (Figure 3c) which according to EV-A71 growth kinetics [47], **ZAF-47** possibly targets EV-A71 post-entry stages like RNA replication and protein translation.

To confirm this finding, assays to investigate the effect of **ZAF-47** on EV-A71 entry into cells were performed. With pre-treatment assay, we first treated the cells with different concentrations of **ZAF-47** before subjecting them to EV-A71 infection. The aim was to investigate if **ZAF-47** was able to bind to host surface receptors which would prevent the virions from binding and thus blocking viral entry. Viral titre was quantified from the supernatant collected 12 hpi and we found that, besides the highest concentration, treatment with **ZAF-47** displayed similar viral titres to the DMSO control (Figure 3d). Even at the highest concentration, the reduction in viral titre (0.8 log_10_ PFU/mL decrease) was not as pronounced as post-treatment and could likely be due to drug toxicity. Thus, it is unlikely that **ZAF-47** inhibits EV-A71 infection by binding with host receptors to block viral entry. With co-treatment assay, we first treated EV-A71 virions with **ZAF-47** for 30 min before infecting the cells. The aim was to investigate if **ZAF-47** was able to bind to viral surface proteins which would prevent the virions from binding to the host receptors and thus blocking viral entry. There was no significant difference between the viral titres of the supernatants collected from **ZAF-47** treatment and DMSO treatment 12 hpi (Figure 3e). Thus, it is unlikely that **ZAF-47** blocks EV-A71 entry into host cells by binding to viral surface proteins. Instead, using entry bypass assay, we were able to deduce that **ZAF-47** most likely targets EV-A71 post-entry stages. With entry bypass assay, we transfected the cells with EV-A71 RNA instead, followed by treatment with different concentrations of **ZAF-47**. The aim was to simulate virus uncoating with transfection and investigate if **ZAF-47** can inhibit EV-A71 post-entry stages after viral genome is released into the cytosol. Unsurprisingly, from the supernatant collected 12 hpi, we found that treatment with **ZAF-47** resulted in significant viral reductions of 0.9 log at 10 µM and 2.6 log at 20 µM (Figure 3f). This finding aligned with previous results and suggests that **ZAF-47** inhibits EV-A71 post entry stages.

### 2.5. **ZAF-47** Reduces Expression of EV-A71 Viral Proteins by Specifically Targeting EV-A71 RNA Replication

Since **ZAF-47** possibly targets EV-A71 post-entry stages, we were interested in looking at the effect of **ZAF-47** on EV-A71 protein expression. To do so, we first infected RD cells with EV-A71. We then treated the infected cells with different concentration of **ZAF-47** for 6 h and collected the cell lysates for SDE-PAGE and Western blot. Analysed with the vehicle control, we found that treatment with **ZAF-47** resulted in significant drop in band intensities of EV-A71 proteins, VP2 (28 kDa) and VP0 (36 kDa). VP2 band could been seen with the vehicle control but not with any **ZAF-47** treatment while VP0 band could be seen with **ZAF-47** treatment at 5 µM but the intensity was reduced by 35-fold in comparison to the control (Figure 4a,b) after normalization with the detected loading control, β-actin. These observations showed that treatment with **ZAF-47** can reduce EV-A71 protein expression in host cells. Since **ZAF-47** acts on the post-entry stages and reduces EV-A71 protein expression, we hypothesized that **ZAF-47** could be inhibiting RNA replication or protein translation, or possibly both, resulting in reduced viral protein expression.

To identify specifically which of these two intrinsically linked processes that **ZAF-47** inhibits, we transfected RD cells with EV-A71 RNA replicons that were either replication-competent or replication-defective and subsequently treated them with different concentrations of **ZAF-47** for 12 h before detecting for luminescence. Both forms of replicons were previously established and validated [48]. The replication-competent replicon is capable of self-replicating as it contains an intact 3D region that encodes for 3D polymerase (3D^pol^) responsible for RNA replication [49] (Figure 4c). Hence, cells containing the replication-competent replicon express luciferase upon IRES translation of the original replicons as well as the replicated copies. In contrast, the replication-defective replicon contains a 3D region with 159 nucleotide deletion which would produce a non-functional 3D protein and impair RNA replication (Figure 4c). As such, cells containing the replication-defective replicon express luciferase upon IRES translation of the original replicons only. The transfected cells were also treated with negative control, DMSO and positive controls, guanidine hydrochloride (GuHCl) and cycloheximide (CHX), a RNA replication-specific inhibitor [50] and a general translation inhibitor [51] respectively. It was observed that **ZAF-47** treatment resulted in significant decrease in luminescence in a dose-dependent manner in cells containing the replication-competent replicon and not in cells containing the replication-defective replicon, similar to GuHCl (Figure 4d,e). This shows that **ZAF-47** is capable in inhibiting viral RNA replication and not IRES translation, indicating that the antiviral activity of **ZAF-47** is specific to EV-A71 RNA replication.

To validate this finding, we also transfected RD cells with a bicistronic reporter construct which had been established and validated previously [52]. Similarly, the transfected cells were treated with **ZAF-47** at various concentrations and then incubated for 12 h before luminescence detection. The construct comprises a human cytomegalovirus (CMV) promoter with a downstream *Renilla* luciferase (R Luc) gene and an EV-A71 strain 41 IRES with a downstream firefly luciferase (F Luc) gene (Figure 4f). This means that R Luc can be expressed by the CMV promoter via cap-dependent translation while F Luc can be expressed by the EV-A71 IRES via cap-independent translation. An accurate determination of IRES activity would be to use the ratio of F Luc luminescence reading to R Luc luminescence reading (F Luc/R Luc). The transfected cells were also treated with negative control, DMSO and positive control, apigenin, an EV-A71 IRES inhibitor [53]. It was observed that **ZAF-47** treatment did not reduce IRES activity in any of the concentration used (Figure 4g). This suggests that **ZAF-47** indeed does not inhibit EV-A71 IRES translation.

### 2.6. **ZAF-47** Possibly Targets Host Factor/s Associated with EV-A71 RNA Replication

EV-A71 was repeatedly passaged in RD cells together with a low initial **ZAF-47** concentration that was gradually increased overtime, to select for resistant mutants that would give us better insight in identifying the possible targets of **ZAF-47** in EV-A71 RNA replication. Unfortunately, we were unable to obtain any resistant mutants as the viral titre for **ZAF-47**-treated remained lower than that of the controls at Passage 21. By infecting RD cells with the supernatant collected from Passage 21 at MOI = 1 and subsequently treating them with **ZAF-47**, we found that **ZAF-47** is still capable of inhibiting the passaged virus dose-dependently (Figure 5). As a result, we hypothesize that **ZAF-47** possibly targets host factor/s associated with EV-A71 RNA replication which makes it harder to generate resistant mutants.

### 2.7. Broad-Spectrum Anti-enterovirus Activity of **ZAF-47**

We were also interested in the antiviral potential of **ZAF-47** against other *Enteroviruses*. As such, we performed post-treatment assay on RD cells infected with other *Enteroviruses* such as CV-A16, CV-A6, Echo7, CV-B5, CV-A24 and EV-D68. **ZAF-47**, at 10 µM, was capable in reducing the viral titres of the above-listed enteroviruses significantly (Figure 6a–f). The respective IC_50_ values upon **ZAF-47** treatment for the different enteroviruses were tabulated in Table 3. The effectiveness of **ZAF-47** against at least four different species of *Enterovirus* suggests that it can function as a broad-spectrum anti-enterovirus agent which is highly useful against HFMD, a common disease caused by different enteroviruses.

## 3. Discussion

In the last two decades, EV-A71 had become endemic across the Asia–Pacific region, causing major outbreaks every few years and resulting in many fatalities and severe neurological complications [33]. Recently, however, it has started to spread out of the region to countries such as the United States [35], Spain [36] and France [37]. With the lack of an internationally approved vaccine or antiviral against these infections and the rapid spread of virus to other parts of the world, there is a crucial need for the development of safe and potent antivirals to combat this persistent global public health problem.

Historically, phenotypic screening has been used for discovery of new drugs. While there has been a paradigm shift to a target-based approach in recent years, many first-in-class drugs with novel modes of action still derived from phenotypic screening [54]. This was shown in our study that by using an immunofluorescence-based phenotypic screen, we were able to identify novel andrographolide derivatives that inhibit EV-A71 infection. Andrographolide is a major constituent of the medicinal plant *Andrographis paniculate* which has an extensive range of therapeutic capabilities including antiviral properties [4]. In China, the sulfonate derivative known as “Xiyanping” is used to treat HFMD with good clinical efficacy [38,39]. Yet, despite its notable pharmacological activities, andrographolide has low water solubility and poor bioavailability [55]; thus, we sought to improve its pharmacology with novel chemical modifications. We successfully synthesized 344 novel andrographolide derivatives and, with the use of a phenotypic screen, we were able to identify 19 anti-EV-A71 hits.

Among the hits, we had previously reported on the anti-EV-A71 activity of AGP-253 [43]; however, we were more interested in **ZAF-46** and **ZAF-47** due to the 8,9-olefin and 14-quinolinoxy modifications they contain which were structurally distinct from AGP-253. From the literature, some 9-dehydro-17-hydro analogues increase neuroprotective properties [56] and are more efficient against angiogenesis [57,58] than their 8,17-olefinic counterpart. These results envision that olefinic transformation from 8,17-olefin to 8,9-olefin is valuable modification direction to discover more potent antivirals based on andrographolide’s antiviral scaffold [38,40,44,59]. In addition, chloroquinoline and hydroxychloroquinoline were reported to be antiviral agents including as ZIKV [60,61] and SARS-CoV-2 [62] inhibitors, and quinoline moiety was broadly used as building blocks in many anti-infectious drug discoveries [63,64,65,66]. From subsequent post-treatment assays, we found that **ZAF-46** was capable of reducing EV-A71 viral titre at 10 µM and 20 µM with a CC_50_ of 29.91 µM and IC_50_ of 4.44 µM while **ZAF-47** was capable of reducing EV-A71 viral titre at 5 µM, 10 µM and 20 µM with a CC_50_ of 29.57 µM and IC_50_ of 2.06 µM. We were intrigued by how the modified derivatives were able to inhibit EV-A71 infection, so we performed mechanistic studies on **ZAF-47**, the more potent of the two compounds (it has a better selectivity index of 14.35).

From the TOA and TOR assays, we found that the antiviral activity of **ZAF-47** acts between 4 hpi to 6 hpi and according to the previous work on EV-A71 viral kinetics [47], this time window seems to correspond to the viral RNA replication and protein translation stages. Nevertheless, we also performed experiments that investigate the effect of **ZAF-47** on the viral entry stages and we found that **ZAF-47** reduces EV-A71 titres in entry bypass assay but not pre-treatment and co-treatment assays which suggests that, indeed, **ZAF-47** acts on EV-A71 post-entry stages. Through the use of Western blot and luciferase assays, we narrowed down the antiviral activity of **ZAF-47** to targeting EV-A71 RNA replication specifically. Besides EV-A71, **ZAF-47** was also able to inhibit other enteroviruses which suggests that it is either acting on a common host factor, since the RD cells were used in all infections, or acting on a viral factor found universally in *Enteroviruses*. The former is likely to be the case as we were unable to generate resistant mutants despite repeatedly passaging EV-A71 with **ZAF-47**. To summarize, we were able to identify novel andrographolide derivatives as EV-A71 inhibitors using an immunofluorescence-based screening approach. **ZAF-47**, a quinolinoxy-andrographolide derivative, was discovered to inhibit EV-A71 RNA replication specifically.

Andrographolide has a multi-targeting nature [67]. One of the targets that it inhibits is the NF-kB signaling pathway shown to be essential for EV-A71 replication and EV-A71-induced inflammatory responses [68,69]. However, it was reported that the increased survival of EV-A71-infected mice using andrographolide sulfonate is due to its immunomodulating effect rather than its antiviral effect [40]. In contrast, andrographolide was able to suppress viral replication of the related EV-D68 by preventing acidification of virus-containing endosomes [18] and our previous work on an epoxy andrographolide derivative was shown to inhibit EV-A71 RNA replication [43]. As such, we believe andrographolide and its derivatives such as **ZAF-47** can have multiple targets in inhibiting EV-A71 infection. Thus, while we hypothesize that **ZAF-47** plausibly targets a host factor responsible for EV-A71 RNA replication, we also do not rule out the possibility of it being a viral factor, especially since andrographolide was reported to have high affinity to SARS-CoV-2 RNA-dependent RNA polymerase (NSP12) [70]. In addition, the quinolinoxy modification in **ZAF-47** could have enhanced its antiviral capability by having a direct or indirect effect on the inhibition of EV-A71 RNA replication. It was reported that some quinolone-based drugs were able to inhibit HIV-1 replication by binding to RNA and interfering with Tat–TAR interaction [64]. By binding to RNA, the quinolinoxy group may have direct inhibition on EV-A71 RNA replication or indirectly by bringing andrographolide close to the site of replication where andrographolide can bind to its target to inhibit EV-A71 RNA replication. Further work needs to be done to identify the specific target(s) of **ZAF-47** in EV-A71 RNA replication.

## 4. Materials and Methods

### 4.1. Synthesis of **ZAF-46** and **ZAF-47**

#### 4.1.1. (14β)-(8′-Quinolinoxy)-9-dehydro-17-hydro-3,19-isopropyleneoxy-andrographolide (**7**)

Under an inert atmosphere, the solution of compound **6** [46] (1 g, 2.56 mmol), triphenylphosphine (PPh_3_) (1.01 g, 3.84 mmol) and 8-hydroxyquinoline (0.56 g, 3.84 mmol) in anhydrous tetrahydrofuran (10mL) was cooled by iced water and then added dropwise diisopropyl azodicarboxylate (0.75 mL, 3.84 mmol). The reaction was monitored by TLC to indicate the reaction was complete in 3 h. The reaction solvent was evaporated, the residue was dissolved and extracted with ethyl acetate, washed twice with saturated aqueous sodium chloride solution, and the organic phase was dried over anhydrous Na_2_SO_4_. The organic phase was evaporated to dryness, and column chromatography was performed to obtain compound **7**: 76.4% yield; white solid; m.p. 72.5–74.3 ^o^C; ^1^H NMR (400 MHz, DMSO-*d*_6_) *δ* 8.89 (dd, *J* = 4.2, 1.8 Hz, 1H), 8.39 (dd, *J* = 8.4, 1.7 Hz, 1H), 7.69 (dd, *J* = 8.4, 1.2 Hz, 1H), 7.62–7.53 (m, 2H), 7.33 (dd, *J* = 7.7, 1.3 Hz, 1H), 6.73 (t, *J* = 6.5 Hz, 1H), 6.14 (d, *J* = 5.4 Hz, 1H), 4.75 (dd, *J* = 10.8, 5.5 Hz, 1H), 4.57–4.48 (m, 1H), 3.85 (d, *J* = 11.6 Hz, 1H), 3.11 (d, *J* = 11.6 Hz, 1H), 2.86 (qd, *J* = 17.8, 6.7 Hz, 2H), 1.94 (s, 2H), 1.83 (s, 1H), 1.67 (s, 2H), 1.56 (s, 1H), 1.45 (s, 3H), 1.31 (s, 3H), 1.26 (s, 3H), 1.21–1.14 (m, 2H), 1.10 (s, 3H), 0.95 (s, 3H), 0.89–0.82 (m, 2H); ^13^C NMR (101 MHz, Chloroform-*d*) *δ* 169.9, 152.5, 151.3, 149.5, 141.5, 136.6, 136.2, 129.9, 128.8, 126.6, 124.4, 122.9,121.8, 116.2, 99.1, 76.1, 74.2, 71.5, 63.9, 48.6, 37.7, 37.7, 33.6, 32.3, 28.9, 26.8, 25.9, 25.5, 24.9, 22.0, 19.5, 18.3. HRMS (ESI) *m/z* 518.2904 [M + H]^+^, calculated for C_32_H_40_NO_5_, 518.2904.

#### 4.1.2. (14β)-(8’-Quinolinoxy)-9-dehydro-17-hydro-andrographolide (**8**)

Compound **7** (0.95 g, 1.84 mmol) was dissolved in 10 mL methanol, and then treated with *p*-toluenesulfonic acid (*p*-TSA) (0.035 g, 0.184 mmol) at room temperature for 1 h. TLC detected reaction progress. After compound **7** consumed completely, the reaction solvent was evaporated and the residue was dissolved and extracted with ethyl acetate, washed twice with saturated aqueous sodium chloride solution, and the organic phase was dried over anhydrous Na_2_SO_4_. The organic phase was filtered, evaporated to dryness, and purified by silica gel column chromatography to afford **8**: 63.8% yield; white solid; m.p. 157.5–159.2 ^o^C; ^1^H NMR (400 MHz, DMSO-*d*_6_) *δ* 8.89 (dd, *J* = 4.2, 1.8 Hz, 1H), 8.39 (dd, *J* = 8.4, 1.7 Hz, 1H), 7.69 (dd, *J* = 8.4, 1.2 Hz, 1H), 7.62–7.53 (m, 2H), 7.33 (dd, *J* = 7.7, 1.3 Hz, 1H), 6.73 (t, *J* = 6.5 Hz, 1H), 6.14 (d, *J* = 5.4 Hz, 1H), 4.75 (dd, *J* = 10.8, 5.5 Hz, 1H), 4.57–4.48 (m, 1H), 3.85 (d, *J* = 11.6 Hz, 1H), 3.11 (d, *J* = 11.6 Hz, 1H), 2.86 (qd, *J* = 17.8, 6.7 Hz, 2H), 1.94 (s, 2H), 1.83 (s, 1H), 1.67 (s, 2H), 1.56 (s, 1H), 1.45 (s, 3H), 1.31 (s, 3H), 1.26 (s, 3H), 1.21–1.14 (m, 2H), 1.10 (s, 3H), 0.95 (s, 3H), 0.8–0.82 (m, 2H); ^13^C NMR (101 MHz, Chloroform-*d*) *δ* 169.9, 152.4, 151.1, 149.5, 141.3, 136.3, 136.1, 129.9, 129.5, 126.6, 124.5, 122.8, 121.9, 115.6, 80.4, 74.0, 71.8, 64.1, 51.7, 42.7, 38.4, 34.8, 34.2, 28.5, 28.0, 22.5, 20.4, 19.4, 18.7; HRMS (ESI) *m/z* 478.2591 [M + H]^+^, calculated for C_29_H_36_NO_5_, 478.2591.

#### 4.1.3. (14β)-(8′-Quinolinoxy)-9-dehydro-17-hydro-19-acetoxy-andrographolide (**ZAF-46**)

To the solution of compound **8** (0.50 g, 1.05 mmol) in dichloromethane (10 mL) and triethylamine (0.36 mL, 2.62 mmol), acetyl chloride (0.15 mL, 2.09 mmol) was added dropwise under ice-water bath. The reaction was complete in 3 h by TLC monitoring. The reaction solvent was evaporated, the residue was dissolved and extracted with ethyl acetate, washed twice with saturated aqueous sodium chloride solution, and the organic phase was dried over anhydrous Na_2_SO_4_. The organic phase was evaporated to dryness, and column chromatography was performed to afford **ZAF-46**: 60.3% yield; white solid; m.p. 83.1–84.9 ^o^C; ^1^H NMR (400 MHz, DMSO-*d*_6_) *δ* 8.89 (dd, *J* = 4.2, 1.7 Hz, 1H), 8.39 (dd, *J* = 8.4, 1.7 Hz, 1H), 7.69 (dd, *J* = 8.3, 1.2 Hz, 1H), 7.62–7.49 (m, 2H), 7.31 (dd, *J* = 7.7, 1.2 Hz, 1H), 6.72 (s, 1H), 6.13 (d, *J* = 5.3 Hz, 1H), 4.76 (dd, *J* = 10.7, 5.5 Hz, 1H), 4.69 (d, *J* = 4.6 Hz, 1H), 4.52 (dd, *J* = 10.8, 1.6 Hz, 1H), 4.11–3.99 (m, 2H), 3.13 (dt, *J* = 10.2, 4.9 Hz, 1H), 2.89 (dd, *J* = 17.9, 7.3 Hz, 1H), 2.77 (dd, *J* = 18.0, 6.0 Hz, 1H), 1.96 (s, 3H), 1.91 (d, *J* = 5.2 Hz, 2H), 1.72 (d, *J* = 12.6 Hz, 2H), 1.58–1.49 (m, 2H), 1.45 (s, 4H), 1.18 (t, *J* = 7.1 Hz, 1H), 1.10 (d, *J* = 12.5 Hz, 1H), 1.03 (s, 3H), 0.69 (s, 3H); ^13^C NMR (101 MHz, Chloroform-*d*) *δ* 171.1, 169.9, 152.4, 151.0, 149.6, 141.4, 136.2, 135.9, 129.9, 129.6, 126.6, 124.6, 122.8, 121.9, 115.8, 78.8, 74.1, 71.4, 65.4, 51.9, 42.1, 38.6, 34.9, 34.4, 28.5, 27.6, 22.4, 21.1, 19.8, 19.4, 19.3; HRMS (ESI) *m/z* 520.2693 [M + H]^+^, calculated for C_31_H_38_NO_6_, 520.2693.

#### 4.1.4. (14β)-(8′-Quinolinoxy)-3-ketone-9-dehydro-17-hydro-19-acetoxy–andrographolide (**ZAF-47**)

The solution of compound **ZAF-46** (0.30 g, 0.58 mmol) in anhydrous dichloromethane (10.0 mL) was treated with Dess Martin Periodinane (DMP) (0.49 g, 1.15 mmol) for about 1 h by TLC monitoring in that the reaction was complete. The residue was dissolved and extracted with ethyl acetate, and then washed with sodium thiosulfate and saturated aqueous sodium chloride solution once, and the organic phase was dried over anhydrous Na_2_SO_4_. The organic phase was evaporated to dryness, and purification was conducted by silica gel column chromatography to provide **ZAF-47**: 75.8% yield; white solid; m.p. 144.5–147.3 ^o^C; ^1^H NMR (400 MHz, DMSO-*d*_6_) *δ* 8.90 (d, *J* = 4.0 Hz, 1H), 8.39 (d, *J* = 8.3 Hz, 1H), 7.69 (d, *J* = 8.2 Hz, 1H), 7.64–7.52 (m, 2H), 7.32 (d, *J* = 7.7 Hz, 1H), 6.73 (s, 1H), 6.14 (s, 1H), 4.77 (dd, *J* = 10.9, 5.6 Hz, 1H), 4.47 (dd, *J* = 38.4, 11.0 Hz, 2H), 3.91 (d, *J* = 11.4 Hz,1H), 2.97 (t, *J* = 8.6 Hz, 2H), 2.64 (d, *J* = 31.9 Hz, 1H), 2.31 (s, 1H), 2.05 (s, 3H), 1.94 (s, 3H), 1.70 (s, 2H), 1.52 (s, 3H), 1.45 (s, 1H), 1.07 (s, 3H), 0.91 (s, 3H), 0.85 (d, *J* = 7.6 Hz, 1H); ^13^C NMR (101 MHz, Chloroform-*d*) δ 213.1, 170.9, 169.7, 152.4, 150.6, 149.6, 141.3, 136.3, 134.9, 130.1, 130.0, 126.6, 124.8, 122.8, 121.9, 115.5, 73.9, 71.3, 65.8, 52.8, 51.1, 38.3, 35.6, 34.9, 33.8, 28.5, 21.4, 20.9, 19.8, 19.7, 19.5; HRMS (ESI) *m/z* 518.2537 [M + H]^+^, calculated for C_31_H_36_NO_6_,518.2537.

### 4.2. Cells and Viruses

This study used the following cells: African green monkey kidney cells (Vero) (ATCC CCL-81) and human muscle rhabdomyosarcoma cells (RD) (ATCC CCL-136). Both cell lines were maintained in Dulbecco’s Modified Eagle’s Medium (DMEM) (Sigma-Aldrich, St. Louis, MO, USA), containing 10% heat-inactivated fetal calf serum (HI-FCS) and 2 g of sodium hydrogen carbonate, at 37 °C with 5% CO_2_. This study used the following viruses: EV-A71 strain 41 (Accession no. AF316321.2); Coxsackievirus A6 (CV-A6) (Accession No. KC866983.1); Coxsackievirus A24 (CV-A24) (Accession No. KF725085.1); Echovirus 7 strain Wallace (Echo7) (Accession no. AF465516); Enterovirus D68 (EV-D68) (Accession No. KM851231); Coxsackievirus A16 (CV-A16) (Accession No. U05876); and Coxsackievirus B5 (CV-B5) (Accession No. JX843811.1). All enteroviruses except EV-D68 (33 °C) were propagated in RD cells at 37 °C with 5% CO_2_ with reduced serum DMEM (2% HI-FCS).

### 4.3. Compound Library

The compound library used in this study was made up of 344 chemically synthesized compounds analogous to andrographolide. A total of 100% DMSO was used to dissolve each compound to attain 10 mM concentration. Compounds were diluted further in serum-free DMEM to attain 100 µM concentration and kept at −20 °C for storage and future use.

### 4.4. Preliminary Screen

Seeded 96-well plates (Corning Inc., Corning, NY, USA), with 2 × 10^4^ RD cells per well, were incubated overnight at 37 °C with 5% CO_2_. The seeded cells were then infected with EV-A71 (50 µL per well, MOI = 1) for 1 h at 37 °C with 5% CO_2_. The EV-A71-infected cells were then treated with the compound library at a final concentration of 10 µM for 12 h at 37 °C with 5% CO_2_. Treatment with 0.1% DMSO was used as vehicle control. Following treatment, cells were fixed with 100 µL of methanol (Sinopharm Chemical, Shanghai, China) for 15 min at −20°C and rinsed with 100 µL of 1x PBS for three times before performing indirect immunofluorescence assay (IFA). Anti-EV-A71 VP2 1° antibodies, MAB979 (Merck Millipore, Burlington, VT, USA), diluted 500-fold in PBS, was incubated with the fixed cells for 1 h at 37 °C. Cells were then rinsed thrice with PBS and incubated with anti-mouse FITC 2° antibodies (Merck Millipore), diluted 200-fold in PBS, for 1 h at 37 °C. After which, the cells were rinsed thrice with PBS before staining with DAPI (Sigma-Aldrich) at room temperature for 15 min. Cells were then rinsed thrice and preserved in 100 µL of PBS before imaging with the Operetta High-Content Imaging System and the Harmony High-Content Imaging and Analysis Software (Perkin Elmer, Waltham, MA, USA). Images of stained cells were captured with the DAPI and FITC fluorescence filters, from a pre-determined central position of each well and processed with the Cell Profiler software [71]. The Cell Profiler software defined the total number of cells and the number of infected cells for each well by the number of fluorescent foci captured from the DAPI-stained nuclei and the FITC-stained EV-A71 respectively. The percentage infection (PI) for each well was calculated using the formula: $\frac{\mathrm{Infected}\;\mathrm{cell}\;\mathrm{count}}{\mathrm{Total}\;\mathrm{cell}\;\mathrm{count}\;}$ × 100%. The percentage inhibition for each well was calculated using the formula $\frac{{\mathrm{PI}}_{0}-\;\mathrm{PI}}{{\mathrm{PI}}_{0}}$ × 100% where PI_0_ refers to the percentage infection of wells treated with the vehicle control while PI refers to the percentage infection of wells treated with the compounds. Compounds that exhibited an inhibition rate of >50% and total cell count comparable to the vehicle control were identified as potential hits.

Before conducting the preliminary screen, the robustness of the preliminary screening assay was assessed with Z-factor. Firstly, 96-well plates, seeded with 2 × 10^4^ RD cells per well, were incubated overnight. 50 µL of EV-A71 (MOI = 1) was used to infect the first 48 wells while the remaining 48 wells were mock-infected with 50 µL of reduced serum DMEM, for 1 h at 37 °C with 5% CO_2_. Following which, additional 50 µL of reduced serum DMEM were added to all the wells for another 12 h at 37 °C with 5% CO_2_ before proceeding to IFA. Percentage infection was derived from IFA images of the wells. Z-factor was then calculated with the following formula: $1-\frac{3\;\left(\sigma_{p}+\sigma_{n}\right)}{\left|\mu_{p}-\mu_{n}\right|}$ where *p* and *n* refer to the positive and negative controls, which refer to EV-A71 infected and mock-infected respectively and μ and σ refer to the mean and standard deviation values of the percentage infection of all wells, either EV-A71-infected or mock-infected, respectively [42].

### 4.5. Cell Viability Assay

To determine the cytotoxicity profiles of the hits, cell viability assay was performed. Seeded 96-well plates, with 2 × 10^4^ RD cells per well, were incubated overnight at 37 °C with 5% CO_2_. Different concentrations of drug hits, diluted by reduced serum DMEM, were used to treat the seeded cells for 12 h at 37 °C with 5% CO_2_. Reduced serum DMEM, was employed as the negative control and 0.1% DMSO as the vehicle control for this step. Subsequently, the supernatant was replaced with alamarBlue^TM^ Cell Viability Reagent (Thermo Fisher Scientific, Waltham, MA, USA) (100 µL), diluted 10-fold in reduced serum DMEM, and incubated for 3 h at 37 °C with 5% CO_2_. Fluorescence intensity in each well was measured at 570 nm excitation wavelength and 600 nm emission wavelength in each well by the Infinite^TM^ 200 series microplate reader (Tecan, Männedorf, Switzerland) and normaliszd with the negative control to determine the relative cell viability.

### 4.6. Post-Treatment Assay

To determine the inhibitory profiles of the hits, post treatment assay was performed. Seeded 24-well plates, with 1.5 × 10^5^ RD cells per well, were incubated overnight at 37 °C with 5% CO_2_. The seeded cells were infected with EV-A71 (100 µL per well, MOI = 1) for 1 h at 37 °C with 5% CO_2_ and then rinsed with 1 mL of PBS twice. Different concentrations of the hits were used to treat the infected cells for 12 h at 37 °C with 5% CO_2_. At 12 hpi, the plates were frozen and thawed for 2 cycles (−80°C; 37 °C) before collection of supernatants to measure the viral titres via viral plaque assay.

### 4.7. Viral Plaque Assays

Measurement of the viral titres from collected supernatants was derived from viral plaque assays. To do this, 24-well plates, seeded with 2.4 × 10^5^ RD cells per well, were used in plaque assays for EV-A71, EV-D68, Echo7, CV-A6, CV-A16 and CV-A24 while 24-well plates, seeded with 1.2 × 10^5^ Vero cells per well, were used in plaque assays for CV-B5. Seeded cells were incubated overnight at 37 °C with 5% CO_2_. Viral supernatants were diluted from 10^−1^ to 10^−7^ in reduced serum DMEM before infecting the cells with a volume of 100 µL for 1 h at 37 °C with 5% CO_2_ (33 °C instead for EV-D68). Cells were rinsed twice with PBS before overlaying with 1 mL of reduced serum DMEM containing 0.5% agarose (Vivantis, Shah Alam, Malaysia). Cells infected with Echo7 supernatants were incubated for 2 days, cells infected with CV-A24 and CV-B5 supernatants were incubated for 3 days and cells infected with EV-A71, CV-A6, CV-A16 and EV-D68 supernatants were incubated for 4 days for the formation of plaques. Subsequently, the cells were fixed and stained overnight with 4% paraformaldehyde and 1% crystal violet (Sigma-Aldrich). Subsequent formation of viral plaques was counted manually to quantify the viral titre in PFU/mL

### 4.8. Time-of-Addition and Time-of-Removal Assays

Seeded 96-well plates, with 2 × 10^4^ RD cells per well, were incubated overnight at 37 °C with 5% CO_2_ before subjecting to EV-A71 infection (50 µL per well, MOI = 1) for 1 h. The infected cells were rinsed twice with PBS. For TOA, cells were incubated with 100 µL of reduced serum DMEM before replacement with 100 µL of 20 µM **ZAF-47** at various time-points of 0, 0.5, 1, 2, 4, 6, 8, 10 hpi. For TOR, 100 µL of 20 µM **ZAF-47** was added to each well, then at similar time-points of 0, 0.5, 1, 2, 4, 6, 8, 10 hpi, replaced with 100 µL of reduced serum DMEM. The plates were frozen down at 12 hpi for viral titre quantification using viral plaque assay.

### 4.9. Pre-Treatment Assay

Seeded 24-well plates, with 1.5 × 10^5^ RD cells per well, were incubated overnight at 37 °C with 5% CO_2_. Cells were treated with 0.1% DMSO or varying concentrations of **ZAF-47** for 2 h at 37 °C with 5% CO_2_. Treated cells were then rinsed with PBS twice before subjecting to EV-A71 infection (100 µL per well, MOI = 1) for 1 h. The infected cells were then rinsed twice with PBS before addition on 1 mL of reduced serum DMEM to each well and further incubated for 12 h at 37 °C with 5% CO_2_. The plates were frozen down at 12 hpi for viral titre quantification using viral plaque assay.

### 4.10. Co-Treatment Assay

Seeded 24-well plates, with 1.5 × 10^5^ RD cells per well, were incubated overnight at 37 °C with 5% CO_2_. EV-A71 virions were incubated with 20 µM of **ZAF-47** or 0.1% DMSO in 1:1 ratio for 30 min at 37 °C before filtration using a 100,000-molecular-weight centrifugal filter unit (Sartorius) for 5 min at 1,500 x g and 4 °C to eliminate unbound molecules. The treated virions were further washed with PBS and filtered again before resuspension in reduced serum DMEM (assuming no loss of virions). The seeded cells were infected with the treated virions (100 µL per well, MOI = 1) for 1 h at 37 °C with 5% CO_2_, rinsed with 1 mL of PBS twice and incubated further for 12 h at 37 °C with 5% CO_2_ after the addition of 1 mL of reduced serum DMEM. The plates were frozen down at 12 hpi for viral titre quantification using viral plaque assay.

### 4.11. Entry Bypass Assay

Seeded 24-well plates, with 1.5 × 10^5^ RD cells per well, were incubated overnight at 37 °C with 5% CO_2_. QIAamp Viral RNA Mini Kit (QIAGEN, Hilden, Germany) was used to extract the EV-A71 viral RNA. In summary, the virions were lysed with viral lysis buffer before precipitating nucleic acids with 100% ethanol for spin-column based extraction. Serum-free DMEM was used to dilute EV-A71 RNA (500 ng) and DharmaFECT 1 transfection reagent (1 µL) (Thermo Fisher Scientific) separately to a final volume of 50 µL before combining them to form transfection complexes. After 20 min of complexing, the seeded cells were transfected with 100 µL of the EV-A71 RNA-DharmaFECT 1 transfection complexes for 1 h. After which, 400 µL of diluted **ZAF-47** or DMSO were added and the cells were further incubated for 12 h at 37 °C with 5% CO_2_. The plates were frozen down at 12 hpi for viral titre quantification using viral plaque assay.

### 4.12. SDS-PAGE and Western Blot

Seeded 24-well plates, with 1.5 × 10^5^ RD cells per well, were incubated overnight at 37 °C with 5% CO_2_. After which, the cells were subjected to EV-A71 infection (100 µL per well, MOI = 1) for 1 h. Reduced serum DMEM was used for mock-infection. The infected cells were then rinsed with PBS twice before treatment with 0.1% DMSO or **ZAF-47** at various concentrations for 6 h at 37 °C with 5% CO_2_. After 6 h of drug treatment, cells were lysed with 100 µL of 1x Laemmli buffer and cell lysate was stored at −80 °C.

The cell lysate samples were subsequently boiled for 10 min at 100 °C and loaded on to a 10% acrylamide gel. SDS-PAGE was run for 2.5 h at 100V to separate the proteins. Bio Basic 10–250 kDa Protein Ladder BZ0011G (Bio Basic, Markham, Canada) was also loaded as a molecular-weight size marker. The separated proteins within the gel were transferred to a polyvinylidene difluoride (PVDF) membrane that was previously activated with methanol, using the Trans-Blot Turbo system (Bio-Rad) for 10 min at 1.3 A. Following which, bovine serum albumin (BSA) (Sigma Aldrich) that was dissolved to 2% concentration in Tris-buffered saline-Tween 20 (TBST) was used to block the PVDF membrane for 30 min. Mouse anti-EV-A71 VP2 1° antibodies, MAB979, was diluted in 2% BSA by 10,000-fold before incubating with the membrane overnight at 4 °C. TBST was used to wash the membrane thrice, 5 min each wash, before 1 h incubation at room temperature with horseradish peroxidase (HRP) conjugated goat anti-mouse IgG 2° antibodies antibody (Thermo Fisher Scientific) that was diluted in 2% BSA by 100,000-folds. After, the membrane was washed thrice with TBST before immersing in Immobilon Western Chemiluminescent HRP substrate (Merck Millipore) for 3 min for detection of protein bands with the C-DiGit Chemiluminescence Western Blot Scanner (LI-COR, Lincoln, NE, USA).

Restore PLUS stripping buffer (Thermo Fisher Scientific) was used to dislodge the antibodies bound to the membrane to re-probe the membrane for β-actin so as to serve as loading control. Similarly, blocking of membrane was performed first before incubation with diluted mouse anti-β-actin 1° antibodies (Merck Millipore) and diluted HRP-conjugated goat anti-mouse IgG 2° antibodies (Thermo Fisher Scientific) for 45 min at room temperature. Both antibodies were diluted by 10,000 fold in 2% BSA. Finally, TBST was used to wash the membrane before chemiluminescent detection of protein bands.

### 4.13. Nano-Luciferase Reporter Assay

Seeded 96-well plates, with 2 × 10^4^ RD cells per well, were incubated overnight at 37 °C with 5% CO_2_. As described in our previous work [48], EV-A71 RNA replicons that were either replication-competent or replication-defective was used to transfect the seeded cells for 4 h at 37 °C with 5% CO_2_. The transfection mixture per well was prepared as followed: EV-A71 replicon (100 ng); DharmaFECT 1 transfection reagent (0.2 µL) and serum-free DMEM (0.2 µL) for complexing before topping up with DMEM; 10% HI-FCS (80 µL). After transfection, cells were washed and treated with 0.1% DMSO, series of **ZAF-47** concentrations, 1 mM GuHCl or 10 µg/mL CHX for 12 h at 37 °C with 5% CO_2_. Nano-Glo kit (Promega, Madison, WI, USA) was used for luciferase detection after 12 h incubation.

### 4.14. Bicistronic Luciferase Reporter Assay

Seeded 96-well white plates (Corning), with 2 × 10^4^ RD cells per well, were incubated overnight at 37 °C with 5% CO_2_. After which, a bicstronic luciferase reporter construct was used to transfect the cells, as described in our previous work [52], for 12 h at 37 °C with 5% CO_2_. The transfection mixture per well was prepared as followed: bicistronic construct (200 ng), jetPRIME transfection reagent (0.4 µL) (Polypus transfection, Illkirch-Graffenstaden, France) and jetPRIME buffer (9.6 µL) for complexing before topping up with reduced serum DMEM (90 μL). After transfection, the cells were treated with 0.1% DMSO, series of **ZAF-47** concentrations or 25 µM of apigenin for 12 h at 37 °C with 5% CO_2_. Dual-Glo Luciferase Assay System (Promega) was used for luciferase detection after 12 h incubation.

### 4.15. Generation of Resistant Mutant

Seeded 24-well plates, with 1.5 × 10^5^ RD cells per well, were incubated overnight at 37 °C with 5% CO_2_. After which, the cells were subjected to EV-A71 infection (200 µL per well, MOI = 1) for 1 h before treatment with 800 µL of **ZAF-47** at final concentration of 10 µM for 12 h at 37 °C with 5% CO_2_. Treatment with reduced serum DMEM and 0.1% DMSO functioned as negative and vehicle controls respectively. Upon 12 hpi, the supernatant was collected for viral plaque assay as well as to infect (200 µL) another seeded plate. This process was repeated until the viral titre for **ZAF-47** treated was comparable to the controls which may indicate generation of resistant mutants.

### 4.16. Antiviral Spectrum of ZAF-47

Post-treatment assays with **ZAF-47** were performed against other *Enteroviruses* (CV-A6, CV-A24, CVB-5, EV-D68, CV-A16 and Echo7) to investigate for antiviral activity. Virus infections were carried out at MOI = 1. **ZAF-47** treatment was performed for 12 h for all enteroviruses except for CV-A16 and CV-A4 (16 h and 4 d respectively). Cell viability assay was carried out to identify the low cytotoxic concentrations for post-treatment assay on enteroviruses with different incubation times (CV-A16 and CV-A6).

### 4.17. Statistical Analysis

Data assessment for statistical significance was performed with one-way analysis of variance (ANOVA) was performed followed by a Dunnett’s post hoc test. Samples that were statistically different from the control had *p*-values < 0.05, 0.01 and 0.001. Two-tailed students’ T-test was performed to compare two samples.

## 5. Conclusions

In conclusion, using an immunofluorescence-based phenotypic screen, we identified novel andrographolide derivatives as EV-A71 inhibitors. Among these hits, **ZAF-47**, a quinolinoxy-andrographolide, was discovered as an EV-A71 RNA replication-specific inhibitor. **ZAF-47** was also effective against other enteroviruses, suggesting a broad-spectrum anti-enterovirus capability that would be very useful in developing antiviral treatment against HFMD.

## Data Availability

The data are contained within the article.

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
