# Peer review of "Discovery of Novel Andrographolide Derivatives as Antiviral Inhibitors against Human Enterovirus A71"

_pharmaceuticals, 2022, doi:10.3390/ph15020115_

Round 1

Reviewer 1 Report

The MS by Tan et al. describes two new andrographolide derivates as antiviral inhibitors against human enterovirus A71. The content is appropriate for Pharmaceuticals, the topic is interesting; however, the manuscript will be accepted after minor revision.

I have the comment that authors have to address to improve the MS:

  • Line 119 -136, For validation of EV-471 hits, The author only focused on ZAF 46 and ZAF 47, but in Figure 2a AGP-49 and AGP-253 displayed cell viability higher than 80%, also; in table 1, the inhibition rate of AGP-253 and AGP 49 higher than ZAF-47? Is the author only interesting in SAR of quinolinoxy and olefin? Furthermore, the other type of compounds should be interesting to explain for SAR. Please describe in detail.
  • Did the author also test andrographolide for EV-471 in this work? I presume only the derivatives of andrographolide. I think it will be interesting we have the data for andrographolide also.

Reviewer 2 Report

Andrographis paniculata (AP)  is one of the prominent medicinal plants in treating many diseases, including viral diseases. In this study, the authors synthesized andrographolide, a major compound of AP, derivatives and tested them against EV-71 which is crucial for the development of anti-viral drugs since there are no approved drugs yet for EV-71. The authors have designed the study correctly and produced data using the appropriate methodology. Overall, the manuscript sounds good.

However, I have encountered that the authors have published a similar paper this month using a different derivative of andrographolide against the same virus (https://www.sciencedirect.com/science/article/pii/S0006295221004366) and the similarity of the contents is high (38% including self-plagiarisms). Otherwise, the findings and data presentation were fine. Therefore, my recommendation is to rewrite the contents free from self-plagiarism too, particularly the introduction, the majority part of the results and the full section of methodology before further processing the paper. Regarding the methods, authors are advised to refer to their published paper and briefly talk about the methods rather than repeating the content in this paper. Additionally, authors need to cite recent papers that discussed the potential application of Andrographis paniculata, especially antiviral effects.

Round 2

Reviewer 2 Report

The authors made substantial corrections. But I could not check the similarity index as the authors did not supply the cleaned file of the revised version. However, there are a few corrections still needed. For example,

  1. The citation for AP is not recent. The author cited 2007, 2014 papers (Ref. 1, 2). There are huge recent papers on AP. Check for other information too. 
  2. English sentence started with the number for multiple sentences. The authors need to change it.
  3. The authors did not cite their own published papers appropriately. Specifically need to refer to what sections were covered from their published paper and what technique they followed.  
  4. "Since the first large outbreak in 1997 in Malaysia, EV-A71 infections have brought about many fatalities and severe neurological complications in the Asia-Pacific region" line 523 mislead the information. EV-A71 outbreak did not start in Malaysia. The authors need to reword the sentence appropriately. 
  5. Line 539-545, the information provided does not fit with the discussion. It seems like the background of the study. Authors need to compress the information and fit it with their findings to make it comprehensive. 
